# Developmentally appropriate mental health literacy content for school-aged children and adolescents

Anna Kågström[1,2] ⬤, Laura Juríková[1,3] ⬤ and Zoe Guerrero[1,2] ⬤

[1]Department of Public Mental Health, National Institute of Mental Health, Klecany, Czech Republic; [2]WHO Collaborating Center for Public Mental Health Research and Service Development, National Institute of Mental Health, Klecany, Czech Republic and [3]Department of Psychology, Faculty of Arts, Charles University, Prague, Czech Republic

## Overview Review

**Keywords:**
Mental health; education; child development; mental health literacy; schools

**Corresponding author:**
Anna Kågström;
Email: Anna.Kagstrom@nudz.cz

## Abstract

Although improving the mental health of children and adolescents has become a global priority, resources outlining developmentally appropriate content for improving mental health literacy (MHL) across school-aged children are scarce. A comprehensive, life-course approach to building MHL is needed to address the evolving competencies, needs, capacities, and risk factors for mental health, especially to establish school-based interventions that can be equitably and sustainably implemented. We conducted a theoretical review highlighting the relation of research and practice in building MHL through developmentally appropriate knowledge and competencies for children and adolescents. A two-pronged review of the literature was conducted to provide an overview of (1) research with a focus on neurobiological, psychological, cognitive, and social developmental milestones of school-aged children relevant for building MHL and (2) evidence-based and theory-driven content for the development of universal MHL interventions for children and adolescents considering the four components of MHL. A map of relevant key milestones highlights the range of development that occurs and ample opportunity for increasing universal MHL during these sensitive years primed for learning. We reflect on current understandings and global considerations for MHL interventions with an emphasis on applying developmental science to the future strengthening of intervention development, uptake, adaptation, implementation, evaluation, and scale-up.

## Impact statement

This review addresses the need for more developmental perspectives linking research to practice within the evidence base of mental health literacy (MHL) interventions for children and adolescents. MHL interventions are foundational to a life-course approach for the prevention and promotion of mental health; however, little research outlines developmentally appropriate content for such interventions in children and adolescents. This review identifies key developmental milestones that are necessary for developing knowledge and skills related to building positive MHL, understanding mental illness and treatments, addressing stigma, and promoting help-seeking for mental health problems. We provide a broad overview of a multidisciplinary perspective to MHL, reviewing empirically based content, mapping developmental milestones, and offering theory-driven knowledge and competencies to be considered within the context of developing and implementing interventions for school-aged children. Follow-up studies are warranted to test the theoretical constructs outlined in this review through the application of developmentally targeted interventions across targeted and universal populations of children and adolescents in diverse contexts around the world. This review sheds light on the need for a life course approach to building MHL and calls for more systematic and comprehensive approaches to building MHL over various developmental and life stages, starting with a robust foundation in childhood and adolescence to promote mental health and well-being.





## Introduction

The pursuit of mental health and well-being is a fundamental right of all, and improving mental health literacy (MHL) positively impacts both individual and population health (Kelly et al., 2007; Kutcher et al., 2016). The burden of mental illness is growing and emerging as a global priority (Vigo et al., 2016), and life-course mental health prevention, promotion, and pathways to care are necessary, starting in childhood and adolescence (Patel et al., 2007). Treatment gaps persist for children and adolescents with mental health problems partially stemming from public stigma and lack of knowledge surrounding symptoms of mental illness which can act as barriers to effective help-seeking (Gulliver et al., 2010).

MHL was originally defined as 'knowledge and beliefs about mental disorders which aid their recognition, management, or prevention' (Jorm et al., 1997, p. 182). This definition has evolved holistically within the field of public mental health to include four components: (1) understanding how to obtain and maintain positive mental health, (2) understanding mental disorders and their treatments, (3) decreasing stigma related to mental disorders, and (4) enhancing help-seeking (Kutcher et al., 2016). This updated definition addresses three interrelated concepts: knowledge, attitudes, and help-seeking efficacy (Wei et al., 2015; Seedaket et al., 2020) and shifted the focus from a mental ill-health approach (or mental disorder literacy) to include the aspects of positive MHL, which has long-term benefits over the life course (Mansfield et al., 2021; WHO and UNICEF, 2021; WHO Regional Office for the Eastern Mediterranean, 2021). MHL is associated with improved community recognition of mental illness and improved attitudes and intended behaviors toward people with mental illnesses (Mcluckie et al., 2014; Milin et al., 2016), as well as improvements in social skills and attitudes toward both help-giving (Lo et al., 2018) and help-seeking (Gulliver et al., 2010; Mumbauer Pisano and Barden, 2020) for mental health problems. Young people especially need to be positively equipped with knowledge, skills, and resources that meet their developmental stages and needs (Renwick et al., 2022).

The building blocks for mental health knowledge and competencies and related health consequences develop early in life, for example, prosocial behaviors and learning skills are associated with decreased health risk behaviors in middle adolescence (Rougeaux et al., 2020). Early adolescence is an especially critical period when considering life course trajectories for the onset of mental illness (Merikangas et al., 2010; Solmi et al., 2022), with 50% of all mental disorders emerging by age 14, and 75% by age 25 (Kessler et al., 2007; Kieling et al., 2011). All children and adolescents should be equipped with knowledge and competencies for mental health self-care, help-seeking, and help-giving behavior to promote their mental resilience and respond to problems appropriately when they arise.

While applications of developmental theory to educational interventions have been long studied and implemented successfully (Smith, 1987), they have yet to emerge within the field of MHL. There is a growing body of evidence indicating the effectiveness of MHL interventions (Amado-Rodríguez et al., 2022), but resources outlining developmentally appropriate MHL content for school-aged children lack (Kutcher and Wei, 2020). Additionally, the links between puberty and brain development with mental health behaviors and outcomes are not well known, in part due to complex and nuanced neurobiological maturation processes that are requisite to the development of new cognitive capabilities (Vijayakumar et al., 2018).

To the best of our knowledge, there is no research overviewing theoretical underpinnings of the developmental appropriateness of specific MHL content. A life-course theory-driven approach to building MHL that takes into consideration the plasticity of the brain and developmental stages across childhood and adolescence could enhance approaches of interventions to meet evolving competencies, needs, capacities, and risk factors for mental health in critical early life stages. MHL interventions should be context-specific, developmentally appropriate, and integrated into existing platforms such as schools or communities (Kutcher et al., 2016). Since primary and secondary schools are optimally positioned to deliver universal equitable and widespread interventions around the globe to improve MHL (Kutcher and Wei, 2020;

WHO, 2021), this review focuses on school-aged children and adolescents.

Due to the broad nature of the topic of development across childhood and adolescence as well as the four complex components of MHL, we draw from a wide range of disciplines and fields to provide an overview of both theoretical and intervention-based research. This review is not systematic and therefore has inherent limitations, notably the difficulty in replicability; however, traditional and systematic literature search strategies did not yield the range of results necessary to cover development nor MHL sufficiently. Therefore, this review should be considered as a foundational overview for meaningful engagement in developmental theory-driven approaches to improving MHL content development, intervention, and testing. First, we reviewed research with a focus on biological, neurological psychological, cognitive, emotional, and social development across school-aged children aged 5–19. Next, we reviewed evidence-based interventions across the four components of MHL and triangulated these with key developmental milestones to construct key knowledge and competencies to be built over the course of childhood and adolescence.

## Developmental milestones of school-aged children and adolescents

The transition from childhood and adolescence is marked by significant neurobiological, physical, psychological, cognitive, and social developments. The behavioral and neurobiological roots of cognition, emotion, and executive functions in adolescence is well known to be a complex transitional phase in life in terms of vulnerabilities (Crone, 2009). When understanding common age patterns in brain activation and behavior, it is essential to keep in mind individual differences (Steinberg, 2005); however, while every child has a unique developmental process, there are general milestones in terms of brain development and associated capacities for learning which can be used as a theoretical foundation for evidence-informed developmental approaches to building MHL.

The prefrontal cortex (PFC), which is responsible for higher cognitive functions (Casey et al., 2000), plays a critical role in the cognition, emotions, and behaviors of emerging adolescents (Yurgelun-Todd, 2007). The development of the PFC also forms the foundation for increased self-awareness (Ochsner, 2004). The maximum density of gray matter is reached first in the primary sensorimotor cortex, later in the PFC (Konrad et al., 2013). Therefore, children and adolescents face difficulties regulating emotions since the structures associated with emotional responses such as the amygdala and ventral striatum mature earlier than the PFC which is involved in cognitive control and executive functioning (Martin and Ochsner, 2016).

While cortical white matter increases throughout childhood and adolescence (Gogtay et al., 2004), gray matter conforms to an inverted U-shaped developmental trajectory. Gray matter initially increases in volume during childhood, reaches a peak in adolescence, and declines steadily into adulthood (Blakemore et al., 2010). The development of the PFC is protracted in comparison to other brain regions, and the interaction of individual regions and their imbalance is associated with heightened vulnerability for engaging in risky behavior, reward-seeking, and emotional reactivity, all of which contribute to increased susceptibility to the motivational properties driving novelty-seeking (e.g., substance use) during adolescence (Casey and Jones, 2010; Sturman and Moghaddam, 2011).

The PFC and the parietal cortex are responsible for executive functions such as goal planning, working memory, decision-making, and selective attention, although different aspects of executive function can have inconsistent developmental trajectories (Blakemore and Choudhury, 2006). Varied rates for the development of distinct cognitive control processes are partially accounted for by differences in maturational time course between prefrontal subregions (Crone et al., 2006). The late maturation of corticostriatal connectivity is thought to support developmental changes in value-guided and goal-directed behavior in adolescents (Insel et al., 2017).

The dorsolateral, ventrolateral, and ventromedial regions of the PFC are also responsible for emotional development, mainly emotional processing, and regulation (Ochsner and Gross, 2008). Prefrontal maturation is required to support complex behaviors, and since it is the last brain region to mature (Huttenlocher and Dabholkar, 1997), guided behavior, working memory, and organizational skills do not reach full capacity until middle-to-late adolescence (Sowell et al., 2002; Luna et al., 2004; Best and Miller, 2010; Caballero et al., 2016). Low emotional regulation is related to youth psychopathology and is associated with many mental disorders (Ahmed et al., 2015). Emotional regulation requires high-level executive and social processes, including working memory, inhibitory control, abstract thought, decision-making, and perspective-taking (Somerville and Casey, 2010; Blakemore and Robbins, 2012; Dumontheil, 2014). These processes are underpinned by structural and functional changes in brain regions that are involved in affect generation and regulation, such as the limbic system and PFC (Ahmed et al., 2015; Martin and Ochsner, 2016). Many brain structures and functions progressively mature and develop throughout adolescence and continue to grow in adulthood (Somerville, 2016).

During childhood and adolescence, the above changes in brain structure and function gradually improve cognitive and impulse control abilities (Bunge and Wright, 2007). From 5 to 9 years old, children progressively move from a self-centered perspective to begin to understand that others can have different feelings or views than their own, develop self-competence and a growing attention span, and learn to recognize and respond to the emotions of others (Selman, 2003). By age 6, children start developing independence and autonomy and begin to form opinions about moral values and attitudes by adopting a sense of right and wrong (WHO, 2021). Children 7–12 years old develop self-reflective perspective-taking and can consider their own and others' opinions and feelings simultaneously, can describe personal experiences in detail, and begin to take interest in what others think and do as the importance of friends is heightened (Selman, 2003). Perspective-taking is crucial in these years for successful social communication and includes awareness of one's own mental state and the ability to ascribe mental states to others (Blakemore and Choudhury, 2006). Higher-order thinking (e.g., problem-solving and self-control) is not fully mature in these early years (Casey et al., 2008).

By the ages of 6–9, children explore different coping mechanisms including developing problem-solving and more complex distraction techniques such as shifting focus (Zimmer Gembeck and Skinner, 2011). By the ages of 6–12, children can effectively communicate their own opinion and understand the emotions and intentions of others (Westby and Robinson, 2014). Children aged 8–12 exhibit lower engagement of the dorsolateral PFC, which reflects the maturation of an additional neural circuit aiding in the performance of higher cognitive tasks and abilities (Bunge and Wright, 2007). Around the age of 9, children can assess their own

thoughts and develop self-confidence (Kleitman et al., 2012), a sense of responsibility and control (Such and Walker, 2004), and are able to express their emotions through words more effectively (ASHA, 2022). Until around the age of 12, children primarily perceive the world via concrete thoughts (e.g., black-and-white thinking patterns). Usually, children younger than this age are more egocentric and self-focused, and therefore can act insensitively to others (Martin and Sokol, 2011).

Around 10 years old, people develop abilities to predict, understand, and conceptualize various perspectives (Selman, 2003), and because of improved long-term memory capacity, they succeed in sequencing, ordering, and classifying information by the age of 11 (Forsberg et al., 2021). Around 11–12 years old, there is a peak of gray matter in the frontal and parietal lobes (Giedd et al., 1999) followed by a sharp acceleration in the loss of gray matter in the dorsal PFC and the parietal cortex (Sowell et al., 2001). This allows for more efficient cognitive processing (Blakemore, 2008), enabling more thoughtful and complex communication via longer and more complicated sentences and engagement in active listening.

From 10 to 14 years old onward, young adolescents can rely on a wider range of problem-solving skills and are capable of better identifying their own inner emotional states and reacting with new self-regulatory coping strategies (Zimmer Gembeck and Skinner, 2011). Early adolescents aged 11–14 experience increased fatigue and need for sleep related to the physical changes of a growth spurt, can experience irritability and secretiveness, are easily embarrassed, and have restricted ability for complex and abstract thought (Gutgesell and Payne, 2004). Though self-control generally increases during adolescence (Zondervan Zwijnenburg et al., 2020), adolescents can be inclined to focus more on the present than the long-term effects and consequences of their actions (Steinberg, 2010).

From the age of 12 onward, youth gradually shift from concrete to abstract thought (Meschke et al., 2012), which enables them to interpret information differently, think complexly, have new and unique thoughts and opinions, and begin to challenge society's expectations (Napolitano et al., 2021). Young adolescents tend to show greater impulsivity and reward-seeking (Steinberg, 2010), experience mood changes (Weinstein et al., 2007), feel more stressed with increased school and domestic responsibilities (Shaw et al., 1996), and their ability to initiate problem-solving grows (Ellis and Siegler, 1994). The imbalance of specific brain regions' maturation peaks during adolescence may contribute to mood instability and greater emotional reactivity in this age group (McRae et al., 2012). Specifically, the limbic and reward systems mature earlier than areas responsible for higher cognitive functions, which causes heightened emotional sensitivity (Konrad et al., 2013).

Adolescents around 12–14 years old experience the peak of amygdala volumes (Goddings et al., 2014). As a result, they become more independent and often challenge, belittle, or ignore adult authority as they gain autonomy (Renk et al., 2005). Emotional independence is one of the core developmental tasks of 12–18-year-old adolescents (WHO, 2021). They can grow fully aware of their own emotions, understand emotions expressed by other people, express their feelings, and self-regulate appropriate emotional responses (Harold and Hay, 2005). Rapid growth within brain regions leads to increased white matter, which is essential for impulse conduction and inhibition, and for learning high cognitive functions (Sowell et al., 1999; Crone et al., 2004), which sets the foundation for strategic thinking. Due to the vast changes brought on by puberty, by 13–14 years old, adolescents are on their

way to establishing their own identities, are more self-aware and self-reflective than prepubescent children as they develop mental flexibility, and develop the capacity for multidimensionality (e.g., switching back and forth between multiple tasks).

Adolescence is characterized by immature PFC activities and enhanced responses in subcortical neural systems, which are important for emotional responses; this can lead to an immature capacity for affective regulation and self-control (Hare et al., 2008). Around 15-years-old, adolescents begin to understand and analyze the causes and effects of emotions (WHO, 2021), and are much more effectively able to regulate their emotions in comparison with 12–13-year-olds (Theurel and Gentaz, 2018). Adolescents from 15 to 17 years old have an advanced cognitive ability due to a decline in gray matter density related to synaptic pruning (Selemon, 2013), so they can think abstractly, multidimensionally, intentionally, and hypothetically. Additionally, 15–17-year-olds understand relationships and associated expectations as well as their roles in society (Gutgesell and Payne, 2004). From the age of 16, self-esteem increases, which carries to middle adulthood (Orth and Robins, 2014) where identity is also more fully formed, specifically self-expression, personal views, and opinions (Waterman, 1982; Koepke and Denissen, 2012).

The reorganization of cortical circuits in adolescence is reflected in the changes in cognitive functions and affect regulation (Casey et al., 2008). From 13 to 17 years old, adolescents strive for maximal independence, and highly value their peers' opinions (Lawler and Nixon, 2011). They are generally motivated by short-term benefits leading to more risky behavior than tendencies of risk avoidance, as those benefits (e.g., social approval) are perceived greater than the risk itself (Romer, 2003). The seeking of more extreme incentives is in part understood as a compensation mechanism for low recruitment of motivational brain circuitry (Bjork et al., 2004). From ages 13 to 16, risky decisions are increased in the presence of peers (Gardner and Steinberg, 2005; Galvan et al., 2007), and sensitivity and perception of negative emotions are heightened (Thomas et al., 2007). During the ages of 13–15, adolescents gain regulation effectiveness as they shift toward a reappraisal strategy and away from, for example, distraction (Theurel and Gentaz, 2018). For older adolescents aged 17+, social connectedness and peer support can drive health-promoting behaviors (Roach, 2018).

## Evidence-based and theory-driven content for child and adolescent mental health literacy interventions

Effective MHL interventions are often described as one or a couple of the components of MHL, touching on topics relating to information on mental health disorders in general or specific mental illnesses such as depression or anxiety, and where students learned how to access help and gained understandings of the lived experiences of peers with mental health problems (Mohammadi et al., 2020; Seedaket et al., 2020; Frețian et al., 2021; Olyani et al., 2021; Patafio et al., 2021). This section provides a brief overview of effective interventions respective to each of the four components of MHL.

There are several types of interventions focusing on the first component of MHL (understanding how to obtain and maintain good mental health). A review of mental health promotion identified five fundamental strategies, including socioemotional learning (SEL), interventions rooted in positive psychology, literacy-based intervention, mindfulness-based interventions, and interventions from the field of positive youth development (Santre, 2022).

SEL interventions teaching emotional processes, social and interpersonal skills, and cognitive regulation are widely implemented and aim to build self-awareness, self-management, social awareness, relationship skills, and responsible decision-making, all of which set the groundwork for promoting mental health (Durlak et al., 2011; CASEL, 2013; CASEL, 2015). Positive Youth Development programs that are focused on increasing a sense of purpose, leadership skills, and self-esteem as well as Positive Psychology Interventions that target strengthening positive emotions through daily routines have also proved effective (Clarke et al., 2021). Finally, mindfulness interventions such as meditation, breath awareness, or yoga practices in which participants can consciously connect with their emotions and feelings are most frequently implemented in school settings mostly at the high school level with adolescents (Zenner et al., 2014; Clarke et al., 2021).

Interventions focusing on the second component of MHL (understanding mental disorders and their treatment) can be divided between general and diagnosis-specific interventions. General interventions cover several mental health disorders in general and are not targeted at any specific mental illness, whereas diagnosis-specific interventions focus on one mental health disorder and can be focused on children and adolescents at risk of developing the specified mental health disorder (Corrieri et al., 2014). Fazel et al. (2014) report on a more complex tiered approach where schools provide universal strategies for all students, followed by interventions to assist selected students at risk, gradually followed by specific treatment interventions for those with greater needs. Diagnosis-specific interventions are most frequently reported within the context of low-intensity interventions such as cognitive behavioral therapy, which are then tailored to be delivered to a given target audience with a specific mental illness diagnosis (Neil and Christensen, 2009; Werner Seidler et al., 2017). Most of such interventions are aimed at older children and adolescents to function as prevention for children and adolescents at risk, usually to reach people with emerging symptoms of depression or anxiety but may also include prevention of risk factors and risk behavior such as substance misuse (Fazel et al., 2014). Understanding mental disorders and their treatments for young children should consider their concrete thinking, and therefore refrain from introducing any complex or nuanced concepts related to mental illness. As children move toward developmental milestones associated with puberty and more abstract thought processes and emotional regulation skills, they become equipped to learn about complex concepts and types of illnesses and associated causes and treatments.

Stigma reduction interventions are most frequently performed in school settings via education-based approaches such as interactive lectures and social contact with a person with lived experience (Hartog et al., 2020). Most interventions targeting stigma have been delivered to older adolescents, and the interventions are often adapted from evidence-based content and approaches for adults rather than being catered to children and adolescents specifically (Heary et al., 2017). Stigma interventions targeting childhood and early adolescence are uncommon, or in cases where children are the focus, the target content and audience focused on the general population and public stigma rather than stigma in children themselves (Hartog et al., 2020). Decreasing stigma toward mental illness requires a range of skills including active listening and empathy, which grow in children over time through practice. Exposure to inclusive environments and practicing these foundational competencies in early years will contribute to normalizing the range of experiences of mental health and mental illness and the recovery process and demystifying misconceptions contributing to stigma.

The final component of MHL (enhancing help-seeking efficacy) is frequently geared toward adults in the lives of children and adolescents rather than children and adolescents themselves (Xu et al., 2018). While there is some understanding of help-seeking facilitators and barriers for children and adolescents, help-seeking efficacy is complex, and the impact of MHL interventions on actual help-seeking is not well understood. Most of the literature links help-seeking with intention rather than actual behavior due to its difficulty to capture via measurement tools (Kelly et al., 2007; Gulliver et al., 2010; Ratnayake and Hyde, 2019). Since emotional regulation is one of the facilitators of help-seeking (Rickwood et al., 2005), and stigma is a well-known barrier (Gulliver et al., 2010), the first three components of MHL undoubtedly contribute to improving help-seeking efficacy. Help-seeking is related to developmental processes, and different sources of help are preferred and utilized across childhood and adolescence. When young people do seek help, they are likely to do so from people whom they know and trust, such as their friends or family (Rickwood et al., 2005; Kelly et al., 2007; Gulliver et al., 2010). However, to engage in appropriate help-seeking behaviors, adolescents must first be able to recognize the problem, acknowledge their need for help, and have positive attitudes toward help-seeking and treatment (Gulliver et al., 2010; Skre et al., 2013). As children approach the age of 9, they become more selective with whom they reach out to for help (Zimmer Gembeck and Skinner, 2011). Bale et al. (2020) suggest that simply knowing whom to seek help from is not enough to support help-seeking behaviors, and young people will benefit from learning about articulating their problems to trusted adults.

## Implications of developmental milestones on the capacity of children and adolescents to build mental health literacy

Comprehensive MHL covering all four components is lacking especially for young children (Kutcher and Wei, 2020). However, children are already developing core skills and knowledge early on which contribute to their mental well-being over the life course and are capable of learning aspects across the four components of MHL. The neurological, psychological cognitive, and social developmental milestones have direct implications on the capacities necessary for building MHL. This section synthesizes and links the aforementioned milestones with developmentally appropriate knowledge and competencies respective to the capacity for understanding and applying knowledge and skills related to building positive MHL, understanding mental illnesses and their treatments, reducing stigma, and enhancing help-seeking according to three age clusters: 5–9-year-olds, 10–14-year-olds, and 15–19-year-olds. In relation to the developmental milestones reviewed above, Figure 1 maps key milestones that are essential for building MHL (see 'Developmental milestones relevant for MHL' in Figure 1). Taking into consideration the state of evidence for child and adolescent MHL interventions and drawing from the theoretical constructs of developmental milestones reviewed in the previous sections, in this section, we extrapolate developmentally appropriate content for MHL across school-aged children and adolescents. Figure 1 further synthesizes and presents developmentally appropriate content in the form of knowledge and competencies to be built over the course of childhood and adolescence following the developmental trajectory (see 'Developmentally appropriate MHL content' in Figure 1).

Children from 5 to 9 years old move from a self-centered perspective as their capacity for self-awareness increases, and they slowly engage in basic perspective-taking (Selman, 2003; Casey et al., 2008). MHL interventions with this age group could tap into the natural growing ability for building empathy as well as expanding self-awareness of emotional states (e.g., intensity or fluctuations of emotions across a few days). As children start to gain these basic skills for understanding and managing their emotions (Selman, 2003; Westby and Robinson, 2014), they are capable of learning about a wide range of emotions including strong emotions, how they present in their bodies, and how they develop self-regulation basic problem-solving strategies. As primary caregivers of this age group are usually the ones with whom children communicate their needs, developmentally appropriate MHL for 5–9-year-olds should encourage children to express their needs, feelings, and mental health problems not only with primary caregivers, but also with other safe adults. Children should be taught to recognize safe and available sources of help as well as the importance of asking for help, and be modeled healthy communication, emotional regulation, and healthy relationships by the adults in their lives. Their ability to grasp and build upon the holistic concept of health including mental, physical, and social aspects as interrelated concepts becomes possible as they shift away from black-and-white thought processes.

Due to brain maturation, 10- to 14-year-olds experience a great shift from concrete to abstract thinking and mental flexibility, an improvement in information-processing skills, and the ability to retrieve the necessary information to solve new problems (Blakemore, 2008; Meschke et al., 2012). They become more able to acquire the ability to analyze and synthesize knowledge about themselves and their experiences, developing meta-cognition and reflective thinking. Engaging in novelty-seeking or risky behaviors is common; therefore, learning to recognize external factors affecting one's mental health would enhance agency and responsibility as it pertains to mitigating risky behaviors related to mental health problems. With improved cognitive functions, 10–14-year-olds are able to understand the consequences of actions in a self-reflective manner, which is an optimal stage for raising awareness about brain development and its connection to mental health (e.g., the biology of emotions and stress response). Improvements at this stage are seen across emotional clarity and awareness, understanding of complex emotions in self and others, emotional regulation, and intentional self-reflection (Harold and Hay, 2005; Zimmer Gembeck and Skinner, 2011; McRae et al., 2012; Konrad et al., 2013). Factual information about mental health influences and states can be taught at this age to support the development of self-compassion and understanding in youth. MHL interventions should tap into the growing ability to learn more complex emotional regulation and management, build empathy, and deepen self-awareness toward individual mental health status over time.

Youth aged 10–14 years are better in perspective-taking and begin to apply values, ethics, and ideology to behavior (WHO, 2021), which can be optimized in activities aimed at building empathy and diversity, acceptance in their interpersonal relationships, and intentional contact to others with diverse experiences (e.g., people with mental illness). As this age group develops more concrete thought, it is appropriate for them to build literacy surrounding basic prevalence rates for mental illness and etiology, and differences between the concept of mental health problems and mental illness. As youth develop a preference for peers as a primary social group as opposed to caregivers, tension can arise in relationships with primary caregivers (Renk et al., 2005). Building understanding and communication skills for relationships with peers and carers could help this age group to build and maintain healthy

Developmental milestones relevant for mental health literacy

Perceives the world through concrete thought

Develops abstract thinking, understands complex thoughts and opinions

Practice basic self-regulation skills

Understand there are multiple ways to interpret situations, emotions

Able to engage in perspective taking

Understand unique emotional perspective

Showing emotion in appropriate contexts

Increased emotional introspection, can analyse causes of emotions

Understands that people can feel multiple emotions simultaneously

Understand more complex emotions in self and others, mixed feelings

Capable of intentional self-reflection

Can distinguish different intensities for emotions

Can regulate emotions through thought processes

Develops autonomy apart from parents

Primarily communicates needs with primary caregivers

Primarily communicates needs with adults

Primarily communicates needs with peers

Able to communicate experiences, thoughts and feelings

Experiencing mood swings

Engages in novelty seeking

Identify trends in personal health behaviors, and distinguish healthy versus unhealthy habits

Practicing conflict resolution

Exhibits mental flexibility

Aware of locus of control

Self-centered perspective

Developing self-awareness

Apply values, ethics, and ideology to behaviors

More able to retrieve information to solve new problems

Gradually increasing attention span

Begin to plan consciously, coordinate actions, evaluate progress, modify plans

Thinks about short/long term consequences of actions

Age  5  6  7  8  9  10  11  12  13  14  15  16  17  18

Developmentally appropriate mental health literacy content

- Understanding a holistic concept of health
- Demonstrating basic problem-solving
- Understanding and perceiving a various range of emotions as normal
- Practicing identifying and regulating emotions with support from adults
- Learning about tolerance, empathy, acceptance of diversity
- Recognizing a wide range of emotions and their presentation in the body
- Developing self-regulation
- Recognizing sources of mental health support (parents, teachers, mental health professionals, etc) and exposure to local examples
- Recognizing emotional states and fluctuations across a few days (i.e. yesterday, today, and tomorrow).
- Communicating emotions and needs
- Building empathy (practicing perspective taking)
- Recognizing importance of asking adult for help when feeling low
- Identify strong emotions and solutions through appropriate help-seeking
- Demonstrating self-awareness of acute mental health status (i.e. identifying emotions when prompted)
- Having safe adult contacts for help-seeking

- Recognizing external factors affecting one's mental health (e.g. emotions, thoughts, behavior, relationship with self and others healthy lifestyle, environmental factors)
- Practicing and establishing self-care and healthy coping mechanisms (e.g. positive self-talk and thinking patterns, basic breathing techniques, enjoyable activities, mindfulness, exercise, relaxation)
- Understanding principles of healthy relationships (e.g. vulnerability, honesty, reliability and trust)
- Linking brain development to mental health (e.g. biology of emotions, hormones, stress response)
- Demonstrating self-awareness of mental health status over time (e.g. changes in mood, behavior, thoughts, lifestyle, stressors)
- Understanding distinction between mental health problems vs mental illnesses
- Knowledge of basic prevalence for mental illness and basic etiology
- Recognizing broad signs and symptoms of mental health problems (non-diagnosis specific) and when to seek help (e.g. when signs and symptoms impact feeling and functioning over time)
- Perspective taking and empathy building for a range of mental health states
- Communicating symptoms of personal mental health problems
- Develop empathetic understanding of experiences of people with mental illnesses
- Knowledge of available support in local community, trusted adults, school resources, helplines
- Understanding the roles of mental health professionals and principle of confidentiality within mental health care
- Understand mental health problems often require support and mental illnesses are treatable (principle of recovery)
- Responding to barriers of help-seeking by trying alternative solutions
- Learning an appropriate and safe role in help-giving for peers in mental distress or disclosing mental health problems (e.g. never promising secrets; supportive communication, active listening, engaging with an adult when needed)

- Setting and maintaining short and long-term strategies and goals for mental health and self-care
- Understanding universal protective and risk factors for mental health across the lifecourse
- Learning basic knowledge of common mental illnesses (e.g. depression, anxiety, etc.)
- Learning about different treatment options and their principles (e.g. psychotherapy, pharmacotherapy)
- Understanding differences in mental health problems requiring self-help, informal help and formal help
- Increasing knowledge about human brain development
- Being self-aware of mental health status over time and identifying contributing factors
- Self-identifying personal risk and protective factors for mental well-being
- Engage in health promoting habits and decrease negative coping mechanisms
- Communicating specific symptoms when help-seeking and associated stressors
- Persistent self-advocating when help-seeking (e.g. anticipating and actively overcoming barriers)
- Ability to recognize mental distress in others and reacting appropriately (e.g. support in seeking help, conversation)

**Figure 1.** Developmentally appropriate mental health literacy content according to foundational milestones across childhood and adolescence.

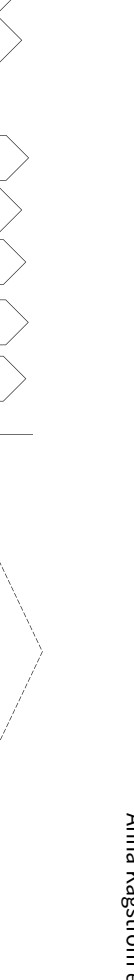

Anna Kågström *et al.*

support networks. Since this age group is at increased risk for the onset of mental health problems, MHL interventions should prepare young adolescents for help-seeking across multiple sources available in respective contexts. While peer-to-peer support is increasingly preferred by growing adolescents, they are also more sensitive to social feedback and peer rejection (Somerville, 2013); therefore, equipping youth to appropriately respond to help-seeking from a peer is also optimal at this age since one negative experience could deter a peer from asking for help again. MHL interventions should teach safety related to help-giving for peers, and issues related to effective support-giving and dangers related to secrecy (e.g., active listening and not promising total confidentiality). Appropriate MHL interventions for this group can focus on establishing self-care strategies and healthy coping mechanisms (e.g., positive self-talk, relaxation strategies, principles of healthy communication, enjoyable activities, and supporting relationships) as well as recognizing broad signs and symptoms of mental health problems and specific help-seeking strategies. Knowledge about the different roles of mental health professionals and the principle of confidentiality within mental health care are important concepts to demystify and promote help-seeking at this age.

Late adolescents, aged from 15 to 19 years, have more mature executive functions (e.g., cognitive flexibility and abstract reasoning) that are necessary for the management of behavior (e.g., self-monitoring and self-control) (Sowell et al., 1999; Crone et al., 2004). Adolescents show a higher capacity for mentalizing (Poznyak et al., 2019), and more autonomy with a higher probability of increased risk-taking, but they are also better at divergent thinking (Cousijn et al., 2014) and exhibit goal-directed behavior and long-term plans for the future. Appropriate MHL interventions for this age, therefore, can focus on setting and maintaining short- and long-term strategies for mental health and supporting healthy habits and self-care. The maturation of the PFC is linked with a greater capacity for self-reflectiveness and self-regulation (Theurel and Gentaz, 2018). These changes, including increased emotional introspection, can create internal distress as young people tend to experience both negative and positive emotions at a heightened state. MHL intervention should focus on building knowledge and promoting self-awareness and self-compassion especially as it relates to brain development, common mental illnesses (such as depression or anxiety), different types of treatments available for mental health care, and the understanding of risk and protective factors for mental health across the life course. As adolescents build independence, gaining competencies for help-seeking is important, including the ability to recognize mental distress in oneself and others, appropriately communicate symptoms of mental distress with others, and crisis management skills (e.g., strategies for activation of the parasympathetic nervous system). Increased empathy and understanding toward unique emotional perspectives occur in later adolescence, which results in sensitivity toward others. Therefore, MHL interventions focused on raising awareness of different mental health problems requiring different types of help, and strategies for adolescent peer-to-peer help-giving can be appropriately understood and used for problem-solving as it is related to managing mental health self-care and help-seeking experiences. Communication skills of this age group are more advanced, and therefore it is appropriate to expect older adolescents to effectively learn and discuss self-advocacy as it related to MHL self-care and help-seeking and giving.

The developmental approach requires long-term strategies toward building blocks which build in sequence over the course of early to late childhood and into adolescence and adulthood.

As such, the theoretical content mapped in this review assumes a life-course approach and commitment to building MHL based on developmental theory, and real-world applications must account for and address any deficits in either knowledge or competencies in the later stages if lacking at baseline. Therefore, it is important that the content mapped and presented in Figure 1 be considered only when embedded within understandings of local contexts and environments.

## Global considerations for mental health literacy content across developmental stages

Global variation in development should be considered when interpreting and applying the content outlined in this review, as well as the evidence base on which implications are drawn. The majority of the global data on child and adolescent development come from high-income countries, and children and adolescents from low- and middle-income settings face developmental challenges at increased rates (Clark et al., 2020). Additionally, developmental changes are unique, prompted by biological maturation, and mediated by environmental factors, and cultural and social contexts. Determinants of brain health, for example, include physical and environmental health, safety, and security, learning and social connection, and access to health and social services which vary widely within and across global contexts (WHO, 2022). The development of children and adolescents is multifaceted and is impacted by a wide range of variables spanning preconception across the life course including risk factors (e.g., maternal deprivation, neonatal and birth risks, effects of malnutrition such as stunting, child labor, exploitation and marriage, adolescent pregnancy and exposure to physical, sexual, and psychological violence and neglect) as well as protective factors (e.g., responsive and skilled pre and post-natal care of the mother, social and group support, immunizations, early childhood supportive interventions, school achievements and parity in education, sale learning environments, information, communication, and technology literacy and universal access to sexual and reproductive health) (Clark et al., 2020).

MHL interventions should be developed and adapted within the context and specific needs of beneficiaries (Kutcher et al., 2016). To that end, there are globally relevant considerations that have implications for MHL intervention across all age groups. Notable considerations include developmental variation across sexes, implementation fidelity, language and culture, and contextual and environmental factors. Sex should be considered and acknowledged when building MHL as milestones can vary slightly among females and males. The onset of puberty starts between the ages of 8 and 14 years in females and 9 and 17 years in males (Blakemore et al., 2010), which has implications for the developmental milestones described above. Additionally, some risk factors for mental illness can vary according to sex (Steinhausen and Metzke, 2001) as well as differences between onset and prevalence of mental illness between females and males (Merikangas et al., 2022). Development of skills related to positive mental health and help-seeking can also vary in children and adolescents. For example, females generally develop prosocial behaviors earlier than their male peers (Van der Graaff et al., 2018) and trends in stages of empathy development differ in males and females (Michalska et al., 2013).

In combination with the quality and developmental appropriateness of the content, the fidelity of the implementation of MHL interventions is of vital importance as outcomes of interventions can be drastically altered due to the degree to which the

intervention is delivered as intended (Gearing et al., 2011). Receptivity to interventions is influenced by correct adaptation interventions to the characteristics of students (Rojas-Andrade and Bahamondes, 2019). In pursuit of high fidelity of MHL interventions, building the capacity of professionals delivering MHL to children and adolescents can be done effectively through training and support, for example, to teachers in school settings (Kutcher et al., 2015a; Kutcher and Wei, 2020).

Cultural and environmental contexts and the availability of mental health resources will have implications on understandings related to mental health and illness, stigma, and help-seeking preferences and pathways. Culture and language play an important role in MHL, impacting how people perceive, talk, and think about mental health and therefore how they experience interventions related to such topics (Quintana et al., 2006). It is therefore important to identify local cultural or ethnic explanations and language around mental health and mental health symptoms to adapt and package content appropriately to ensure high-quality delivery and receptivity (Griner and Smith, 2006). Especially important for increasing help-seeking efficacy through MHL is knowledge of local mental health service provision and pathways to care.

## Conclusion

Building and expanding upon evidence-based practices from developmental theory, this overview provides a foundation of theory-driven developmentally appropriate MHL content for children and adolescents to be applied for further definition, application, and testing. There is much yet to learn, and further intervention design, implementation, and evaluation are underway and necessary to explore more emerging evidence within the field of MHL through the lens of development. Exploring the generalizability and universality of translating developmental research to practice will be a key step to better understanding active ingredients and key content for building MHL in children and adolescents moving forward. For MHL intervention development and implementation, understandings of local contexts and cultures, and relevant developmental determinants, risks, and protective factors, must be taken into consideration when adopting or adapting MHL content for interventions. Future research should test the specific developmental applications of MHL to diverse contexts including universal, selected, and targeted interventions, and explore the trajectories of knowledge and competencies over developmental stages. Continued research on MHL across diverse global settings, including regions with high stigma and low resources, will be important toward validating or challenging developmental aspects of MHL presented in this review.

**Open peer review.** To view the open peer review materials for this article, please visit http://doi.org/10.1017/gmh.2023.16.

**Author contribution.** A.K. initiated and supervised the review. All authors participated in designing the methodology, conducted the literature review, and contributed to writing and proofreading the draft. A.K. and L.J. critically revised the manuscript.

**Financial support.** This work was supported by MH CZ – DRO (National Institute of Mental Health – NIMH, Czech Republic, IN: 00023752) as well as by EEA and Norway Funds Project: SUPREME Strengthening Universal Prevention, Resources, and Evaluation of Mental Health in Education, funded via the grant 'Monitoring a posilování duševního zdraví dětí a adolescentů' (Grant No. ZD-ZDOVA1-025). The funding bodies had no role whatsoever in the design of our program or study, methodology used, data collection, data analysis, data interpretation, or writing of this paper.

**Competing interest.** The authors declare none.

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
