## [Reviewer Report]

Dear Editors,

I am pleased to submit our original commissioned review entitled Developmentally appropriate mental health literacy content for school aged children and adolescents to the Cambridge Prisms: Global Mental Health.

We believe this review addresses the need for more developmental perspectives linking research to practice within the evidence base of mental health literacy interventions for children and adolescents. The study sheds light on the need for a life-course approach to building mental health literacy and calls for more systematic and comprehensive approaches to building mental health literacy over various developmental and life stages.

As mental health literacy grows in practice and research, this article provides a rich foundation for continued global engagement in bringing research to practices in the development and implementation of mental health literacy interventions for school-aged children.

Thank you for your time, and for considering this research article.

Sincerely,

Anna Kagstrom

Anna Kagstrom

Department of Public Mental Health, 

National Institute of Mental Health, Czechia 

Topolová 748

250 67 Klecany 

Czechia,

anna.kagstrom@nudz.cz

---

## [Reviewer Report]

*Comments to Author*: Thank you for the opportunity to review this manuscript. This paper seeks to provide a theoretical review of the developmental course of school aged children (from childhood to adolescence) to inform the development of universal MHL interventions for this target group. This is an interesting review with many practical applications to the development of population level MHL interventions, particularly for school aged children and young people. At the same time, some questions emerged as I read through the review. I hope the below comments are helpful to the authors, as I do believe this is an important and novel paper.

The discussion of developmental changes in childhood and adolescence was very interesting. This section could perhaps benefit from presenting information either by age group (e.g. starting with the developmental milestones in children and working through chronologically to adolescence) or by cognitive skill (e.g. arrange more clearly by topic – ‘emotional regulation’, ‘impulse control’, ‘social relationships’, ‘concrete vs abstract thinking’). At times I found it difficult to follow the threads of different changes that can occur throughout the early lifespan. Figure 1 is an excellent framework in the way it clearly sets out the overlapping but separate foundational milestones.

The section titled ‘Evidence based and theory driven content for child and adolescent mental health literacy interventions’, seemed to focus much more broadly on providing an overview of MHL interventions for school aged children, an area which, while of adjacent interest, has been covered before. Instead, the authors could consider focusing the discussion on the implications of these life course changes for MHL content development, which is one of the main novel contributions of this manuscript. This information is included beautifully in Figure 1 (under ‘Developmentally appropriate mental health literacy content’) and I felt the inclusion of this in the main manuscript would enhance its impact. For example, the discussion on the architecture underpinning the importance of peer relationships for young people has implications for how adolescents seek help and what sources they prefer to seek help from – which would imply that MHL interventions that encourage help-seeking may therefore need to be targeted at enhancing adolescents’ ability to support their peers (peer to peer support).

The global applications proposed by this review suggest that MHL interventions should pay attention to cultural contexts (such as local cultural/ethnic differences), potential developmental differences across sexes and fidelity of implementation. Whilst these are important considerations, are there other considerations that can be directly extrapolated from the conclusions you have made about developmentally appropriate content? 

There are minor grammatical issues (for example, in the abstract, the sentence fragment “...especially to inform interventions for school-based interventions where equitable and sustainable interventions can be implemented.” has the word ‘intervention’ three times) that could be amended easily

---

## [Reviewer Report]

*Comments to Author*: Thank you for the opportunity to review, “Developmentally appropriate mental health literacy content for school aged children and adolescents”. This theoretical review examines research with a focus on social developmental milestones of school-aged children relevant to building, as well as empirical and theoretical research for the development of MHL interventions for young people. This paper could make an important contribution to the literature through overviewing theoretical underpinnings of the developmental appropriateness of MHL interventions for young people. However, I have some concerns about the manuscript which I invite the authors to address. 

In the introduction, the authors could expand more on definitions and conceptualisations of mental health literacy. For example, it may be helpful to provide the original definition of MHL provided by Jorm et al. (1997), given this is the most adopted in the field. More recent conceptualisations of MHL are more holistic in nature, and includes reduced stigma and help-seeking efficacy, as well as understanding how to obtain and maintain positive mental health (Kutcher et al. 2016), and it would be helpful to touch on how the concept of MHL has evolved. It currently feels as if definitions of MHL have been overlooked in the introduction, and a strong grounding in what exactly MHL is would strengthen the opening to the paper. Given that you refer to the “four components of MHL” throughout the paper, I think you need to be very clear that these are from Kutcher and colleagues. You introduce this on page 2 (“MHL includes understanding how to obtain and maintain good mental health, understanding mental disorders and their treatments, decreasing stigma related to mental disorders, and enhancing help-seeking efficacy…”) but it is not immediately clear as you progress through the paper, that these are the 4 components to which you are referring. 

While the section, “Developmental milestones of school-aged children and adolescents” does a good job of summarising cognitive development, sections of it seem extraneous, and should be better linked to the goals of the review. Whilst it is helpful to highlight, for example, that the PFC reaches peak grey matter density later than the sensorimotor cortex, and therefore “children and adolescents face difficulties regulating emotions” how does this link to the success of MHL interventions? Whilst it is helpful to include summaries of development here, it may be kinder to the reader to be more selective about which milestones are summarised, and how they link to appropriateness of MHL interventions. You do this well in the beginning of the subsequent section (“Evidence based and theory driven content for child and adolescent mental health literacy interventions”). 

I am concerned by your lack of methods section – It is important to have more transparency about your search techniques, how many records were returned, which were selected for inclusion in the review, and why. What were exclusion and inclusion criteria? I appreciate that the authors state that this is not a systematic review, but some detail about search methods is vital to include. 

On page 7, the authors state, “While there is some understanding of help-seeking facilitators and barriers for children and adolescents, help-seeking efficacy is complex, and the impact of MHL interventions on actual help-seeking are not well understood.” There is a large degree of literature which links help-seeking intention (rather than actual behaviour), and this should be acknowledged. For example, Ratnayake & Hyde,2019; Gulliver et al., 2010; Kelly et al., 2007)

You do however, provide good a good summary of global considerations in the development and delivery of MHL content.

---

## [Reviewer Report]

Dear Editors,

I am pleased to submit the revised version of our review entitled Developmentally appropriate mental health literacy content for school aged children and adolescents.

We are grateful for your valuable feedback and the comments we received from the Reviewers. We provided responses to each comment and updated the manuscript accordingly. In addition, we made some grammatical and editing improvements throughout the manuscript.

Thank you for reconsidering this article for publishing.

Yours sincerely,

Anna Kagstrom

---

## [Reviewer Report]

*Comments to Author*: Thank you for the responses to my comments and for the revision of the paper.

Thank you for addressing the section on the implications of these developmental changes for MHL interventions. This section made sense and I felt tied together the purposes of your review nicely. I have realized reading through the paper again that you have primarily focused on cognitive developmental changes, and not more broadly on other factors that may be pertinent to children’s development in particular (such as their relationship with their primary caregiver viz. the bioecological model, although this is touched on page 33 in the section where you mention encouraging children to express their needs to their primary caregiver). The implication from a cognitive perspective is that MHL interventions should target children in a developmentally appropriate way and their understanding of their own mental health; but I think the larger implication of this developmental stage is that MHL interventions also need to be targeted at the primary support systems of children as well to be optimally effective – because children develop within these systems and are reliant on those caregivers to access optimal care. I think this stood out more once you had divided the implications up into age ranges, with the 5-9 age range presented first. I don’t think you necessarily have the space/scope to cover this, but the wider importance of other developmental factors to childhood development is conspicuously absent and may be one of the limitations of the paper that you could note.

The sentence on page 31 regarding Fazel et al. (2014) could potentially be removed unless you have room to expand on the differences and overlaps between MHL interventions and MH prevention programs in schools.

There are some minor grammatical issues throughout that could be easily rectified (e.g. “There is a growing body of evidence indicating the effectiveness of MHL interventions (Amado- Rodríguez et al., 2022) but resources outlining developmentally appropriate content for MHL for school-aged children lack (Kutcher & Wei, 2020).” – would read more clearly to say ‘are lacking’)

---

## [Reviewer Report]

*Comments to Author*: Thank you for taking the time to make such well-considered amendments to your manuscript. 

You’ve taken on board comments from both reviewers and, as a result the manuscript is more coherent. Having read your response, and the revised manuscript, I think this is ready for publication.